# Dosimetric Analysis of the Short-Ranged Particle Emitter ^161^Tb for Radionuclide Therapy of Metastatic Prostate Cancer

**DOI:** 10.3390/cancers13092011

**Published:** 2021-04-22

**Authors:** Peter Bernhardt, Johanna Svensson, Jens Hemmingsson, Nicholas P. van der Meulen, Jan Rijn Zeevaart, Mark W. Konijnenberg, Cristina Müller, Jon Kindblom

**Affiliations:** 1Department of Radiation Medical Sciences, Institution of Clinical Science, Sahlgrenska Academy, University of Gothenburg, 413 45 Gothenburg, Sweden; jens.hemmingsson@phonsa.gu.se; 2Department of Medical Physics and Bioengineering, Sahlgrenska University Hospital, 413 45 Gothenburg, Sweden; 3Department of Oncology, Institution of Clinical Science, Sahlgrenska Academy, University of Gothenburg, 413 45 Gothenburg, Sweden; johanna.b.svensson@vgregion.se (J.S.); jon.kindblom@vgregion.se (J.K.); 4Center for Radiopharmceutical Sciences ETH-PSI-USZ, Paul Scherrer Institute, 5232 Villigen, Switzerland; nick.vandermeulen@psi.ch (N.P.v.d.M.); Cristina.Mueller@psi.ch (C.M.); 5Laboratory of Radiochemistry, Paul Scherrer Institute, 5232 Villigen, Switzerland; 6Radiochemistry, South African Nuclear Energy Corporation (Necsa), Brits 0240, South Africa; janrijn.zeevaart@necsa.co.za; 7Department of Radiology and Nuclear Medicine, Erasmus MC, Rotterdam, The Netherlands; m.konijnenberg@erasmusmc.nl

**Keywords:** dosimetry, prostate cancer, PSMA, ^161^Tb

## Abstract

**Simple Summary:**

A tremendous effort and rapid development of the prostate-specific membrane antigen (PSMA)-targeting radio ligands for radionuclide therapy has resulted in encouraging response rates for advanced prostate cancer. Different radionuclides have been utilized or suggested as suitable candidates. In this study, a dynamic model of metastatic progress was developed and utilized to estimate a radiopharmaceutical’s potential of obtaining metastatic control of advanced prostate cancer. The simulations performed demonstrated the advantage of utilizing radionuclides with short-range particle emission, i.e., alpha-emitters and low-energy electrons. The recently-proposed beta-emitting radionuclide terbium-161 demonstrates great potential of being a future candidate towards targeted radionuclide therapy of advanced prostate cancer. This is in line with recent encouraging preclinical results and development of upscaling the product quality. Recently, the first in-human application with a [^161^Tb]Tb-DOTATOC also demonstrated good SPECT image quality, which can enable dosimetry calculations for new ^161^Tb-based radiopharmaceuticals.

**Abstract:**

The aim of this study was to analyze the required absorbed doses to detectable metastases (D_req_) when using radionuclides with prostate specific membrane antigen (PSMA)-targeting radioligands to achieve a high probability for metastatic control. The Monte Carlo based analysis was performed for the clinically-used radionuclides yttrium-90, iodine-131, lutetium-177, and actinium-225, and the newly-proposed low-energy electron emitter terbium-161. It was demonstrated that metastatic formation rate highly influenced the metastatic distribution. Lower values generated few large detectable metastases, as in the case with oligo metastases, while high values generated a distribution of multiple small detectable metastases, as observed in patients with diffused visualized metastases. With equal number of detectable metastases, the total metastatic volume burden was 4–6 times higher in the oligo metastatic scenario compared to the diffusely visualized scenario. The D_req_ was around 30% higher for the situations with 20 detectable metastases compared to one detectable metastasis. The D_req_ for iodine-131 and yttrium-90 was high (920–3300 Gy). The D_req_ for lutetium-177 was between 560 and 780 Gy and considerably lower D_req_ were obtained for actinium-225 and terbium-161, with 240–330 Gy and 210–280 Gy, respectively. In conclusion, the simulations demonstrated that terbium-161 has the potential for being a more effective targeted radionuclide therapy for metastases using PSMA ligands.

## 1. Introduction

There has been encouraging progress for utilizing the prostate specific membrane antigen (PSMA) towards targeted radionuclide therapy of advanced prostate cancer. The first clinical studies were performed with ^90^Y- or ^177^Lu-labeled monoclonal antibody PSMA-J591 [1]. Since then, great efforts were made to produce small-molecule-based radioligands with more favorable kinetics. This resulted in the clinically-applied PSMA radioligands [^131^I]I-MIP 1095, [^177^Lu]Lu-PSMA-617, [^177^Lu]Lu-PSMA-I&T, [^225^Ac]Ac-PSMA-617, [^213^Bi]Bi-PSMA-617, and [^212^Pb]Pb-PSMA-CA012 [2,3,4,5,6,7,8,9,10,11]. For these ligands, high absorbed doses and tumor-to-normal tissue absorbed doses (TNDs) for the critical organs including the kidneys, bone marrow and salivary glands, have been reported (Table 1).

One factor in the development of optimized radioligands is the selection of the radionuclide. Yttrium-90 has the benefit of emitting electrons of high energy per decay and might overcome inhomogeneous activity distributions by cross-irradiation. For advanced disseminated cancer, the absorbed energy to small tumor cell clusters is, however, too low to enable complete remission [12]. Therefore, electron-emitters with shorter path lengths are used, e.g., iodine-131, or lutetium-177. This intention was further pursued by radiolabeling PSMA ligands with the α-emitters actinium-225 and bismuth-213 [6,7]. While the short half-life of 46 min for bismuth-213 was considered too short for obtaining high TNDs, the half-life of 10 days for actinium-225 generated high TNDs and notable clinical response was observed. In an even more exploratory phase are the studies using [^212^Pb]Pb-PSMA-CA012, as combined alpha and electron-emitter, with [^203^Pb]Pb-PSMA-CA012 for diagnostic imaging and prospective dosimetry [8]. In line with this theory, it is speculated that the short-ranged electron-emitter terbium-161, with its abundant number of conversion and Auger electrons, might be a potential candidate for the treatment of advanced prostate cancer (Table 2) [13,14]. In addition, the superiority of [^161^Tb]Tb-PSMA-617 over [^177^Lu]Lu-PSMA-617 for the treatment of small-cell clusters was recently demonstrated in a preclinical setting [15].

The aim of this study was to analyze the relative therapeutic potential of yttrium-90, iodine-131, lutetium-177, actinium-225, and terbium-161, respectively, for the treatment of disseminated prostate cancer, and to estimate the required absorbed doses to obtain a high probability of metastatic control for different advanced metastatic scenarios. In addition, we investigated whether diffused visualized tumor burden is more challenging to treat than oligo-distributions of metastases.

## 2. Materials and Methods

### 2.1. The Metastatic Dissemination Model

The model used for dissemination of metastases was a modification of an earlier Monte-Carlo-based method [12], where it was assumed that the metastatic formation rate (*MFR_i_*) from a tumor depended on its volume at time t (*V*(*t*)*_i_*):(1)$$MFR_{i}=c_{i}\cdot V{\left(t\right)}_{i}^{0.67}$$
where *c_i_* is the formation rate constant. It was further assumed that both the primary tumor and the metastases gave rise to new metastases and that *c_i_* was equal for all tumors in a specific model simulation. Simulations of metastatic distributions were performed for the *c* values between 0.16 and 2.56%/day/g.

Figure 1 gives a description of the metastatic model, showing that new metastases can also be disseminated into already established metastases [17]. The fusion of metastases can either be a direct deposit of tumor cells into an established metastasis or metastases can grow into each other.

### 2.2. The Tumor Growth Model

The growth rate of the tumors was assumed to deaccelerate during development according to the Gompertzian model:(2)$$V\left(t\right)=V_{0}\cdot e^{\frac{SGR_{0}}{b}\left(1-e^{-bt}\right)}$$
where *V*_0_ is to the initial tumor volume, equal to a single cell with mass 1 ng; *SGR*_0_ is the initial specific growth rate; *b* is the growth retardation factor, which can be determined by measurement of the specific growth rate for detectable tumors *SGR*_d_, i.e., tumors that can be volume determined (*V_d_*):(3)$$b=\frac{SGR_{d}-SGR_{0}}{ln\frac{V_{d}}{V_{0}}}$$

The *SGR_d_* for prostate cancer is often estimated from the increased PSA values and can range between 0 to 2.5%/day [18]. We used the median *SGR_d_* of 0.773%/day for a tumor diameter of 20 mm. The choice of *SGR*_0_ will be determined to the maximal volume that can be reached. The initial growth rate of the tumor was set to 7%/day, i.e., the maximum volume is equal to 65 cm^3^ (diameter of 50 mm).

### 2.3. Absorbed Tumor Doses

The decay data of yttrium-90, iodine-131, lutetium-177, actinium-225, and terbium-161 was obtained from ICRP 107, and the full beta spectra were used in the simulations (Table 2) [16]. The absorbed electron energy fraction into spherical tumors with uniform activity distribution was calculated with the PENELOPE 2018 Monte Carlo code. It was assumed that all α-disintegration of actinium-225 decay into to stable bismuth-209, occurred in the tumor. Furthermore, it was assumed that all radiopharmaceuticals had the same tumor biokinetics, with an effective half-life (*T_eff_*) of 51 h [3]. The absorbed energy in spherical tumors with uniform activity distribution was calculated with an in-house developed Monte Carlo code.

Metastatic prostate cancer is predominantly located in normal tissues with low uptake, e.g., skeleton and lymph nodes; therefore, the cross-irradiation from normal tissue are low [19]. The absorbed doses to the simulated tumors in the metastatic distribution were calculated by:(4)$$D_{T}=\frac{T_{eff}\cdot 1.6\cdot 10^{-13}}{\ln\left(2\right)}\cdot \frac{C_{T}}{TNC}\cdot \left(\underset{j}{\sum }E_{j}k_{j}\phi_{T,j}\right)$$
where *T_eff_* is the effective half-life; *E_j_* is the energy (keV) of the emitted charged particle or photon per decay; *C_T_* is the tumor activity concentration, *TNC* is the tumor-to-normal-tissue activity concentration ratio, *k_j_* is the number of emitted charged particles or photons per decay; *φ_T,j_* is the absorbed energy fraction of the emitted charged particles or photons in the tumors, respectively. The constant 1.6·10^−13^ converts keV/g to Gy.

### 2.4. Biological Effective Dose and Radiosensitivity

To determine the radiosensitivity of prostate cancer for targeted radionuclide therapy, we used the biological effective dose (*BED*) concept, which is the total absorbed dose that would be required to cause a biological effect with low-dose-rate irradiation:(5)BED=D·(1+Tr(Tr+Teff)1αβ)
where *α* and *β* is the radiosensitivity constants in the linear and quadric model, respectively. *D* is the dose; *T_eff_* and *T_r_* is the effective half-life and repair rate of DNA damage, respectively. The radiosensitivity value *α*/*β* of 1.5 Gy (95% CI 1.25–1.76) and repair half-life (*t_r_*) of 1.9 h (h) (95% CI 1.4–2.9 h) was used [20], and the *T_eff_* was set at 51 h (range 14–160 h) [3].

The reported alpha values from clinical and preclinical studies have presented a huge variation from 0.036 to 0.487 Gy^−1^ [21]. In this study, we estimated the *α*-value from the review paper by Zaorsky et al., where they demonstrated that a *BED* over 200 Gy for treatment of local prostate cancer had no further beneficial treatment outcome [22]. By inserting this value into Equation (6), having *TCP* = 0.99, the *α* values for tumor diameters of 50 mm in patients with high risk of metastases will be 0.147 Gy^−1^ [23,24]. This value was used in the calculation, however, we also used a radiosensitivity that was double as high for demonstrating the impact of cellular radiosensitivity on the required dose.

### 2.5. The Metastatic Control Probability Model

The tumor cure probability can be expressed as:(6)$$TCP_{i}={\left(1-e^{-\alpha \cdot BED}\right)}^{h}$$
where *h* is the number of clonogenic tumor cells in tumor *i*. The clonogenetic cell density was set to 10^9^ cm^3^, which is within the wide range as determined by Wang et al., i.e., 5.6 × 10^4^ to 1.3 × 10^9^ and far from the unrealistic values of 10–100, as estimated when the *α*-values have been set extremely low, i.e., 0.04 Gy^−1^ [21].

The metastatic cure, or control, probability is the product of all TCPs for the tumors in the disease:(7)$$MCP=\overset{m}{\underset{i=1}{\prod }}TCP_{i}$$

However, since the tumor uptake and radiosensitivity vary inter- and intra-individually, the *MCP* can be described as the probability to achieve high metastatic control. For the required *C_T_* to obtain *MCP* = 0.99, the required absorbed dose, D_req_, was calculated for local energy deposited from the charged particles. This is an appropriate approximation for tumor sizes over 10 mm in diameter. A Monte Carlo code written in MATLAB was used to simulate the metastatic dissemination. At least 1000 simulations were performed to obtain the D_req_ for one parameter setting.

## 3. Results

Dependent on the tumor growth rate, metastatic formation rate, and time after primary tumor formation, the simulations resulted in different metastatic distributions (Figure 2; Table 3). The c values ranged from 0.16 to 2.56%/day/g. In Figure 2, it is shown that the metastatic distribution has the highest frequencies of large metastases for small c values (Figure 2A) and the opposite distribution for high c values with the highest frequency of small metastases (Figure 2B).

To relate these simulation results to clinically useful data, we defined a limit of detectable metastases equal to 7 mm in diameter [25], and analyzed the metastatic distribution for 1 to 20 detectable metastases (Table 3). A low metastatic formation constant implies the mean size of the detectable metastases is larger than for a high metastatic formation constant. The detectable metastases median sizes for the low metastatic formation rate is between 0.8 and 5.0 cm^3^, and the total number of metastases is relatively low. This indicates that these metastases will be detected as separate solid oligo metastases (Figure 3A–C). In contrast, a high metastatic formation rate will generate small visually detectable metastases (median size: 0.40–0.81 cm^3^) and the total number of metastases will be high, indicating that these metastases will be visualized as diffused gathered small spots on nuclear images (Figure 3D).

The electron emitter yttrium-90 was determined to have high absorbed energy in large spheres, but substantially less absorbed energy for small tumors (Figure 4). However, if the activity concentration is high in surrounding tissue, the cross-irradiation from the normal tissue can sterilize the smallest tumors. Figure 4B illustrates this fact for a TNC of 10, i.e., TCP increase for decreased tumor sizes (<10^−6^ cm^−3^). These small tumors can receive increased absorbed energy/decay with iodine-131, but the energy absorbed in small tumors is increased further when using lutetium-177. An even higher absorbed energy/decay could be obtained with terbium-161, which has the lowest variation in absorbed energy/decay for the various spheres. This property is an explanation of the high TCP values for terbium-161 over a wide range of sphere sizes (Figure 4B).

As soon as metastases are formed, the required absorbed doses differ widely between the studied radionuclides (Table 4). For yttrium-90, the low absorbed energy fraction in small tumors will increase the required absorbed dose to over 3300 Gy. Even with iodine-131, the required absorbed dose is in the high range. Substantially lower absorbed doses are required for lutetium-177, but the results also show that the required absorbed dose had to be increased by about 40% for a cancer with 20 detectable metastases as compared to one detectable metastasis. Using actinium-225, the required equivalent dose, multiplied with a weight of 5, is about half of that of lutetium-177 and, when using terbium-161, the required absorbed dose is further reduced to be between 200 and 300 Gy, depending on the metastatic distribution used.

## 4. Discussion

In this study, the potential of various therapeutic radionuclides to achieve high probability of metastatic control for different metastatic scenarios were described. The analyses were applied to the situation of prostate cancer and radionuclides that are currently used in clinics for radioligand therapy. In addition, terbium-161, which we previously proposed as a promising therapeutic radionuclide, was analyzed due to its unique electron emission pattern with both medium-energy electrons (β^−^-particles) and a high abundance of low-energy electrons (conversion and Auger electrons) [13]. The analysis of this study revealed that the range of the low-energy electrons of terbium-161 will ensure a high absorbed dose, even in micro-metastases, and that the dose theoretically required to achieve complete remission using terbium-161 would be even lower than for α-emitters, such as actinium-225.

The radionuclide terbium-161 has previously been highlighted as a potential radionuclide for targeted radionuclide therapy [13,14,15,26,27,28,29]. The present study complements these previous investigations by demonstrating the high potential of [^161^Tb]Tb-PSMA ligands for metastatic prostate disease. It was demonstrated that the required absorbed dose for metastatic control was less than 40% of what was required for the clinically-used radionuclide lutetium-177. This beneficial value for terbium-161 is due to the high emission of electrons with energies below 40 keV, which causes a high local absorbed dose at the cellular level. Simulation studies have recently demonstrated that the cellular absorbed dose is more than three times higher for terbium-161 than for lutetium-177 [30,31,32]. The ratio decreases with increasing cell cluster size; at a cell cluster diameter of 0.1 mm the ratio is 1.8 and the smallest diameter which results in similar absorbed dose of terbium-161 and lutetium-177 to the cell clusters are about 2 mm. A cell cluster with a diameter of 2 mm contains about 10^6^–10^7^ tumor cells, but is still beyond the detectable size of PET imaging. The high absorbed doses to these undetectable micro-metastases are of critical importance to achieve disease control, and thus the therapeutic potential of terbium-161 is superior to that of lutetium-177.

In the metastatic process of prostate cancers, the preferable site of metastasis growth is the endosteal niche in the bone marrow cavity [33]. Once established in the endosteal, the tumor cells can transform and cause imperfect bone remodeling, which may cause fracture and patient pain. However, the growth rate of infiltrating metastases is not well characterized and might differ from the primary tumor, as well as between individual metastases. Furthermore, once established in the endosteal niche the tumor cells might be in a dormant state for an unknown period [34]. In the constructed metastatic dissemination model, it was assumed that all tumors had the same growth rate, which is a simplification of reality, however, the growth rate distribution is unknown and a mean value should be representative of the dissemination process—with the aim to distinguish different radionuclides’ potential of metastatic control. If dormant small cell clusters are present, it should still be preferable to use short-ranged particle emitters such as terbium-161. In such a scenario, terbium-161 might even be a better choice over more high-energy electron emitters. However, such analyses need to be performed with an adjusted model of metastatic spread.

The calculations are based on uniform uptake in the tumors, since non-uniformity of PSMA-radioligand uptake in highly PSMA-positive tumors has not been described. This situation is further confirmed by the high response rates obtained with the short-ranged α-emitter actinium-225, even in large prostate tumors in the liver, indicating that uniform activity distribution is achieved clinically [6]. Furthermore, complete object remission is obtained, but with relapses of the non-visible metastases, indicating that a major problem is that the radionuclides used generate too low absorbed doses to small tumors [35]. However, if non-uniform distribution occurs, the required absorbed doses will also not be uniform, with the increased risk that some cells will obtain too low absorbed dose to achieve cell kill. In the worst-case scenario, it will impact such that metastatic control, by our definition, will never be obtained. Nevertheless, the treatments with [^177^Lu]Lu-PSMA ligands have already demonstrated beneficial response rates, and the use of terbium-161 has the potential to enhance the treatment effect further.

In this study, clinical data was used to determine the radiosensitivity, which was estimated to be 0.147 Gy^−1^ [22]. With this radiosensitivity, the required absorbed doses will even be high for terbium-161, namely, 210–280 Gy. The reported radiosensitivity for prostate cancer ranges over one order of magnitude from 0.036 to 0.49 Gy^−1^ [21]. However, these values are also estimated with respect to the assumed clonal tumor cell density. In the present study, we assumed a high clonal tumor cell density. However, in the studies that have revealed low radiosensitivity the clonal tumor cell capacity was 10 to 100 cells per cm^−3^. With such low clonal tumor cell capacity, the tumor growth rates would be nearly zero, which is obviously not realistic from a clinical perspective. If a radiosensitivity that is twice our estimated radiosensitivity was applied, the required absorbed dose for terbium-161 would be substantially lower, i.e., 120–170 Gy. These results indicate a potential for combining the use of radiolabeled PSMA agents with available radiosensitizing agents. One example of emerging agents with proposed radiosensitizing properties are the second-generation anti-androgens, such as Enzalutamide, already in clinical use for metastatic castration resistant prostate cancer [36,37,38].

In contrast to terbium-161, the long electron range from yttrium-90 emission makes it an unsuitable alternative for treatment of advanced metastatic spread; compared to terbium-161, the required absorbed dose would be more than 10 times higher in order to obtain metastatic control. The use of iodine-131 is successful in the treatment of hyperthyroidism and well-differentiated thyroid cancer, due to the high natural metabolic uptake mechanism in the thyroid tissues to be treated. However, for labeling of tumor targeting agents with the purpose to treat small metastases, iodine-131 is less suitable than radionuclides with short-ranged electron emission. Furthermore, iodine-131 emits a high quantity of photons that nonspecifically irradiate organs and tissues such as the radiosensitive bone marrow. The clinical study with [^131^I]I-PSMA ligand also addressed the bone marrow toxicity as being problematic [5]. Nevertheless, a Phase II study with [^131^I]I-MIP-1095 is ongoing and might reveal the clinical potential of using ^131^I-based radiopharmaceuticals for aggressive metastatic prostate cancer [39].

The use of ^177^Lu-labeled PSMA ligands demonstrated encouraging early clinical results in treatment of advanced castration-resistant prostate cancer [3,9,40,41,42,43]. The retrospective comparison with the standard of care of third-line treatments showed a clearly improved response rate for [^177^Lu]Lu-PSMA-617; 55% of the patients obtained a best PSA response of more than 50% versus 22% for standard third-line treatments. In addition, fewer side effects were observed with [^177^Lu]Lu-PSMA-617 [2,44] compared to standard therapies, such as cytotoxic chemotherapy. Our results indicate the degree of improved response that could be achieved if radioligand treatments are inserted earlier in the disease than performed for reported clinical studies. A large increase in required dose is needed when metastatic disease is present; i.e., in this analysis, equal to when a metastasis is observed with PET, CT or MRI. The required dose increase is more than twofold and it further increases depending on the number of tumors observed, as well as the metastatic dissemination pattern. The first randomized Phase III clinical trial, VISION, has concluded patient inclusion and first results are expected in the near future [39].

To further improve the response rates by the selection of an adequate radionuclide, researchers at DKFZ, Heidelberg, Germany, employed actinium-225 [6,35]. The decay of actinium-225 is complex and the fate of all daughter radionuclides formed are unsure. However, since PSMA-radioligands are efficiently internalized, it is proposed that the daughter radionuclides will remain intracellular. Nevertheless, the recoil after initial decay will most probably release a number of daughter radionuclides before being internalized. The fate of these radionuclides is unknown and is not included in the modelling, which only takes the internalization of the [^225^Ac]Ac-PSMA-617 into account.

The treatment response of [^225^Ac]Ac-PSMA-617 was similar to that of lutetium-177; however, [^225^Ac]Ac-PSMA-617 was applied to the group of patients with diffused metastases, while [^177^Lu]Lu-PSMA-617 was used for patients with oligometastases. In spite of our results, demonstrating higher required absorbed doses for diffused disseminated tumors, this strategy also stratified the patients into a group more challenging to treat. We demonstrated that not only the number of visible tumors, but also the metastatic pattern, influences the required absorbed dose for obtaining complete remission. Despite the responses achieved with actinium-225 in this patient group, it appears that short-ranged particles has its value in the treatment of metastatic prostate cancer. Today, three ongoing Phase I/II clinical trials are exploring the clinical potential of [^225^Ac]Ac-PSMA-617 and [^225^Ac]Ac-PSMA-J591 [39].

The number of short-ranged particle emitters for patient treatment are scarce [45]. Alpha emitters are generally not easily available [46]. As demonstrated in this study, a valid alternative would be terbium-161. In addition to the advantageous electron emission profile, terbium-161 also emits photons useful for quantification of the absorbed doses to the critical organs using SPECT. Recently, a clinical SPECT protocol was established, demonstrating even higher spatial resolution compared to that of ^177^Lu-based SPECT [47]. In addition, the production of no-carrier-added terbium-161 in large quantities and at a quality comparable to no-carrier-added lutetium-177 is feasible [26,27]. These facts enabled a first-in-human application with [^161^Tb]Tb-DOTATOC in two patients with somatostatin receptor positive tumors [48], which further indicates that ^161^Tb-labeled PSMA ligands might have great potential for clinical application.

## 5. Conclusions

Out of the radionuclides studied our results indicate that [^161^Tb]Tb-PSMA ligands have the highest potential for improving the response rates of advanced metastatic prostate cancers. These results were obtained by applying a dynamic model of the dissemination of metastases. It was demonstrated that both the number of metastases and dissemination pattern affects the required dose necessary for complete remission of the disease. The highest impact on the required dose is, however, given by the radionuclides which emits ideally short-ranged particle radiation rather than long-ranged radiation. In this respect terbium-161—with its emission of abundant low-energetic electrons—is the most promising radionuclide candidate for PSMA-ligand based treatment of metastatic prostate cancer.

## Figures and Tables

**Figure 1 cancers-13-02011-f001:**
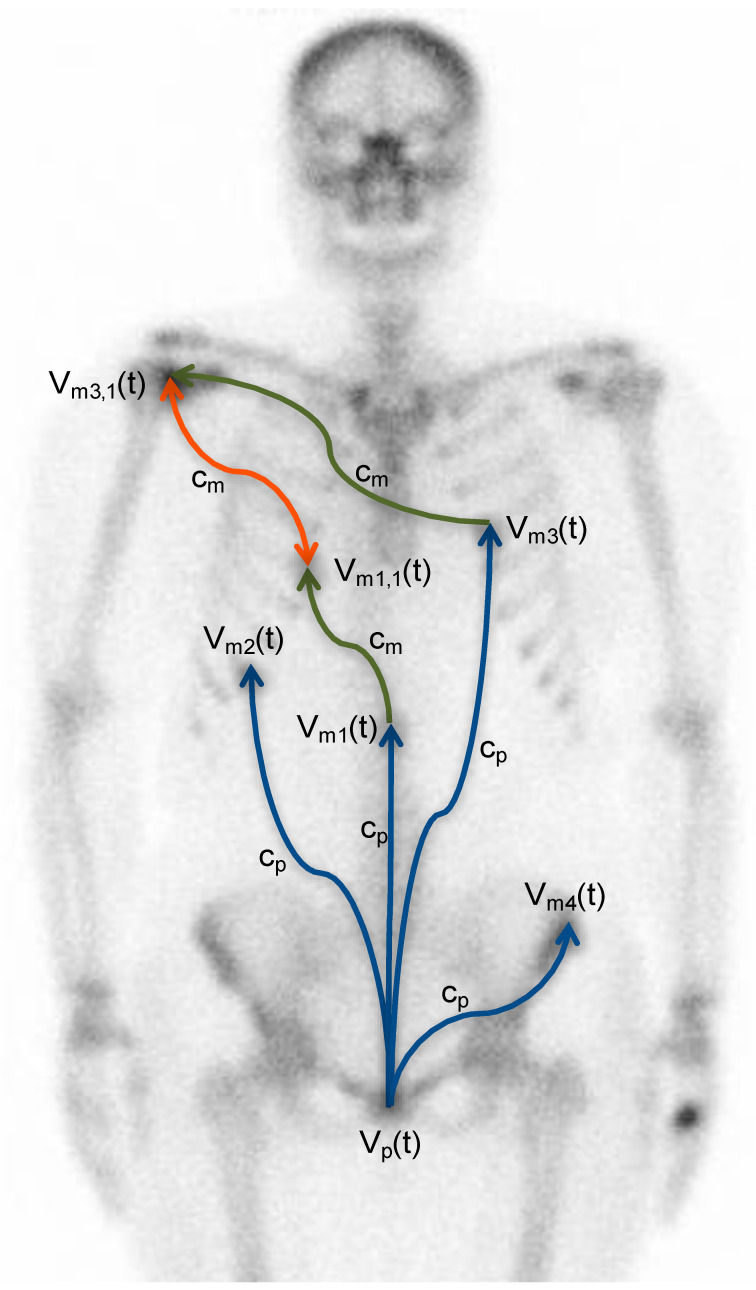
Schematic description of the metastatic dissemination model of prostate cancer. The metastatic formation rate (c_p_) depends on the volume of the primary tumor (V_p_(t)). From the settled metastases (V_mi,i_(t)), new metastases can be formed at a metastatic formation rate c_mi_. The blue arrows illustrate the dissemination of metastases from the primary tumor, while the green arrows illustrate dissemination from the metastases. The orange doubled arrow illustrates that tumor cells can be disseminated in any direction into already-established metastases [17].

**Figure 2 cancers-13-02011-f002:**
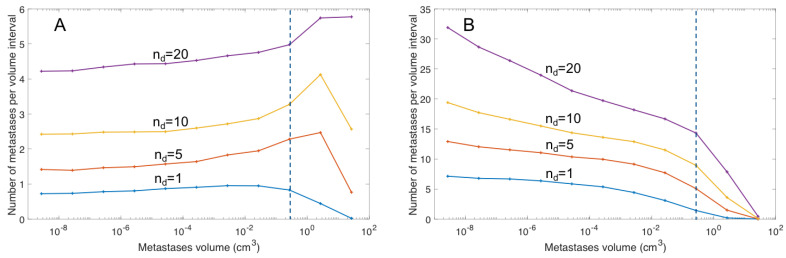
The metastatic distribution for different metastatic formation rates (**A**) c = 0.16% d^−1^g^−1^ (**B**) c = 2.56% day^−1^g^−1^. The dashed blue line indicates the assumed detection limit of metastases, equal to 7 mm. The blue, red, yellow and magenta lines show the metastatic distribution for 1, 5, 10, and 20 detectable metastases (*n*_d_), respectively.

**Figure 3 cancers-13-02011-f003:**
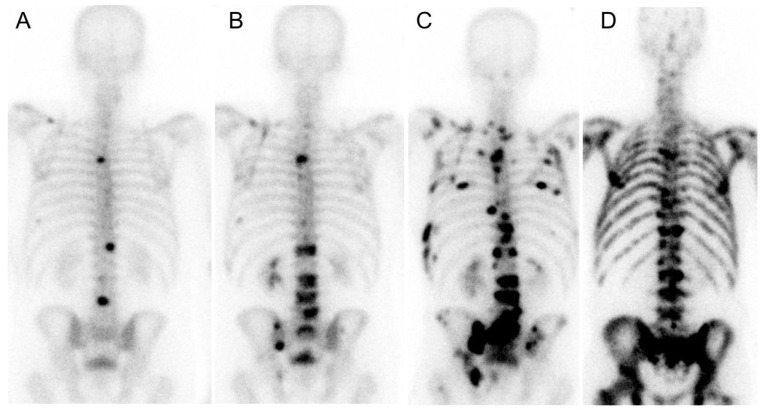
Bone scans performed with [^99m^Tc]Tc-phosphonate in prostate cancer patients. (**A**) Initial patient scan visualizing well-defined oligo metastases. (**B**,**C**) At later time points, 400 days after initial scanning (**B**) and 800 days after initial scanning (**C**), the number of well-defined metastases increased despite continued systemic treatment; (**D**) A typical patient with diffused visualized metastases in the skeleton.

**Figure 4 cancers-13-02011-f004:**
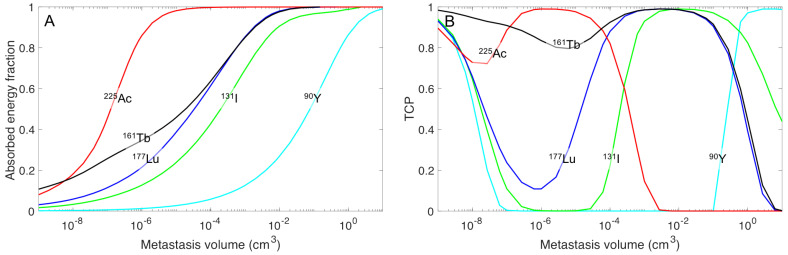
(**A**) The absorbed energy fractions for the emitted charged particles and the tumor cure probability (TCP) for ytterium-90 (cyan), iodine-131 (green), lutetium-177 (blue), terbium-161 (black), and actinium-225 (red) in spherical tumor volumes with homogenous activity distribution. (**B**) Graph shows the TCP curve for the activity concentration that generates an optimal TCP = 0.99, and a TNC = 10, for the sphere masses of 10^−9^ (one cell) to 100 g.

**Table 1 cancers-13-02011-t001:** Absorbed doses per administered activity and TNDs for kidney, bone marrow, and salivary glands after treatment with radiolabeled PSMA ligands.

PSMA-Ligand/ Study	*n*	Range	Tumor (Gy/GBq)	Kidney (Gy/GBq)	Bone Marrow (Gy/GBq)	Salivary Gland (Gy/GBq)
[^177^Lu]Lu-PSMA-617 Baum el at. [3]	8	mean TND	3.3 (0.03–78)	0.8 4.1 (0.03–97)	0.025 132	1.3 2.5
[^177^Lu]Lu-PSMA-617 Fendler et al. [9]	15	mean TND	6.1	0.55 11	0.02 (0.05) 305	1.0 6.1
[^177^Lu]Lu-PSMA-617 Scarpa et al. [10]	10	mean TND	3.4 (1.1–7.17)	0.6 (0.11–1.39) 4.9 (2.2–66)	0.04 (0.02–0.1) 74 (24–290)	1.0 (0.48–2.7) 3.8 (2.0–8.3)
[^177^Lu]Lu-PSMA-617 Violet et al [11].	30	mean	5.3 (0.41–11)	0.39 (0.09–0.84) 14	0.11 (0.01–0.34) 48	0.58 (0.13–1.87) 9.1
[^225^Ac]Ac-PSMA-617 Kratochwil et al. [6]	4	mean TND	5.7 Sv_5_/MBq	0.74 Sv_5_/MBq 7.7	0.05 Sv_5_/MBq 110	2.3 Sv_5_/MBq 2.5
[^213^Bi]Bi-PSMA-617 Kratochwil et al. [7]	3	mean TND	6.3 Sv_5_/MBq	8.1 Sv_5_/MBq 0.78	0.52 Sv_5_/MBq 12	8.1 Sv_5_/MBq 1.3
[^212^Pb]Pb-PSMA-CA012 dos Santos et al., [8]	2	mean TND	140 mSv_5_/MBq	49 ± 2 mSv_5_/MBq 2.9	6.2 ± 1.2 m Sv_5_/MBq 23	75 m Sv_5_/MBq 1.9

Note: The authors use of Sv, with a weight of 5, are applied in Table 1.

**Table 2 cancers-13-02011-t002:** Physical properties of the studied radionuclides [16].

Physical Properties	^90^Y	^131^I	^177^Lu	^161^Tb	^225^Ac/../../../ ^209^Bi
Main decay modes	β-	β-	β-	β-	α/β-
Physical half-life (d)	2.7	8.0	6.6	6.9	10.2
Beta/CE/Auger (keV/decay)					
0.01–10	0	1	2	9	6
10–50	1	6	9	43	18
50–100	2	13	16	16	24
100–200	7	41	52	45	62
200–1000	294	129	67	87	504
1000–9000	630	0	0	0	52
Alpha (keV/decay)	0	0	0	0	5787/6304/7068/5846 (2.1)/8377 (97.9)
Main photon-energies keV (%)	Bremsstrahlung	364 (82) 637 (7)	113 (6) 208 (11)	26 (23) 49 (17) 75 (10)	218 (11) ^221^Fr 440 (26) ^213^Bi

CE: conversion electrons.

**Table 3 cancers-13-02011-t003:** Simulation results of the number of detectable metastases for two different metastatic formation constants. For the median size of detectable metastases, the 95% confidence interval is presented in parentheses.

Number of Detectable Metastases	Metastatic Formation Constant (%/day/g)	Formation Time (d)	Median Size of Detectable Metastases	Total Number of Metastases	Total Metastatic Burden (cm^3^)
1	0.16	1390	0.80 (0.19–9.0)	7.70	1.80
5	0.16	1750	2.0 (0.20–23.8)	18.7	23.8
10	0.16	1990	3.5 (0.21–35.1)	33.6	80.2
20	0.16	2290	5.0 (0.21–46.7)	67.1	228
1	2.56	1070	0.40 (0.18–3.0)	47.3	0.94
5	2.56	1220	0.54 (0.19–5.0)	91.3	5.72
10	2.56	1310	0.66 (0.19–7.2)	136	14.7
20	2.56	1410	0.81 (0.19–9.7)	220	37.7

**Table 4 cancers-13-02011-t004:** The range of the required absorbed tumor dose for obtaining MCP = 0.99 at different metastatic formation rates and its corresponding number of detectable metastases. The radiosensitivity α was set to 0.147 and TNC set to 100. For actinium-225, a weight of 5 was used in the calculation of the required absorbed dose; indicated with the unit “Gy_5_”.

Radionuclide		Required Absorbed Dose			
		Number of Detectable Metastases			
	0	1	5	10	20
^90^Y	80 Gy	2630–3010 Gy	2740–3130 Gy	2860–3220 Gy	3000–3330 Gy
^131^I	80 Gy	920–1120 Gy	985–1170 Gy	1040–1220 Gy	1100–1270 Gy
^177^Lu	80 Gy	558–682 Gy	598–715 Gy	630–742 Gy	672–777 Gy
^225^Ac	80 Gy	243–292 Gy_5_	252–307 Gy_5_	269–320 Gy_5_	288–335 Gy_5_
^161^Tb	80 Gy	207–247 Gy	220–260 Gy	230–270 Gy	245–281 Gy

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
