# Peer review of "Dosimetric Analysis of the Short-Ranged Particle Emitter 161Tb for Radionuclide Therapy of Metastatic Prostate Cancer"

_cancers, 2021, doi:10.3390/cancers13092011_

Round 1
Reviewer 1 Report
This manuscript provided an interesting comparison using Monte Carlo dosimetry between PSMA peptides labelled with a range of radionuclides, including Tb-161in their ability to treat prostate cancer metastases grown at different rates.
Minor comments:
- Nomenclature needs to be changed according to https://pubmed.ncbi.nlm.nih.gov/29074076/
- Abstract: The abstract and main text of the manuscript mention the Dreq for different radiopharmaceuticals; it is unclear however whether this Dreq is whole body, i.e. all tumour mets together, or each met on its own (more than likely the first, but do make this clear).
- Abstract: final sentence: change to “being a more effective targeted radionuclide therapy for metastases..”
- Change “Auger-electrons” to “Auger electrons”
- Line 73 “small cell clusters” not “small cell cluster”
- Table 2: explain what nt is
- Should “c” in the main text be “ci”?
- Line 128: change “preferably” to “preferentially”
- Line 129: insert space to “lymphnodes”
- Add to discussion whether modelling assumes a lack of recoil from alpha particles even before they get to tumour sites?
- What is the beta value set at?
- What were the alpha, beta, Teff etc based on?
- Line 155: change to “probability” not “probable”
- Line 158: change to “clonogenic” not “clongenetic”
- Line 166: change to “described” not “described”
- MATLAB code, add where this code will be accessible? Github?
- Introduce acronyms when first used, including d for day.
- Figure 2 is confusing, why the dip?
- Table 3: is the mean size of mets an average of each met individually or in of all mets combined? Also, if there is a mean, please do add an SD value.
- Table 4: for Ac-225, why is there an Gy5in each column rather than a Gy written down?
- Add in methods or discussion what percentage binding is assumed of PSMA, which PSMA peptide is being used for the assumptions, what the assumed specific activities are, and whether an equal affinity is assumed for the target regardless of which chelator or radionuclide is used (this won’t always be the same).
Author Response
This manuscript provided an interesting comparison using Monte Carlo dosimetry between PSMA peptides labelled with a range of radionuclides, including Tb-161in their ability to treat prostate cancer metastases grown at different rates.
Reply: Thank you for your interest in our study. Below we give our response to your supporting comments.
Minor comments:
- Nomenclature needs to be changed according to https://pubmed.ncbi.nlm.nih.gov/29074076/
Reply: We have now changed the nomenclature according to the reference.
- Abstract: The abstract and main text of the manuscript mention the Dreq for different radiopharmaceuticals; it is unclear however whether this Dreq is whole body, i.e. all tumour mets together, or each met on its own (more than likely the first, but do make this clear).
Reply: This has now been clarified in the abstract, and more details are described in the method sections. The following changes have been inserted in abstract (line 29-30) :
The aim of this study was to analyze the required absorbed doses to detectable metastases (Dreq) when…
- Abstract: final sentence: change to “being a more effective targeted radionuclide therapy for metastases.”
Reply: Thanks, now corrected.
- Change “Auger-electrons” to “Auger electrons”
Reply: Changed.
- Line 73 “small cell clusters” not “small cell cluster”
Reply: Corrected.
- Table 2: explain what nt is
Reply: nt has been changed to decay
- Should “c” in the main text be “ci”?
Reply: It should be acceptable to use c in the main text, since we stated that all ci was set to be equal for all for all metastases in a simulation.
- Line 128: change “preferably” to “preferentially”
Reply: Corrected.
- Line 129: insert space to “lymphnodes.
Reply: Corrected.
- Add to discussion whether modelling assumes a lack of recoil from alpha particles even before they get to tumour sites?
Reply: Such a situation will decrease the tumor uptake, but it will not affect the performed calculations. The following has been inserted in the discussion (line 528-531):
“Nevertheless, the recoil after initial decay will most probably release a number of daughter radionuclides before being internalized. The fate of these radionuclides is unknown and is not included in the modelling, which only takes the internalization of the 225Ac-PSMA-617 into account.”
- What is the beta value set at?
Reply: The beta value does not have to be specified, since we use the alpha/beta ratio of 1.5 and determine an alpha value, as described in the method.
- What were the alpha, beta, Teff etc based on?
Reply: These values were based on analysis of clinical external irradiation of primary prostate cancer, performed by other groups (the references are given in the method section). Teff was determined from biokinetic studies in patients. The reference is provided in the method section.
- Line 155: change to “probability” not “probable”
Reply: Corrected.
- Line 158: change to “clonogenic” not “clongenetic”
Reply: Corrected
- Line 166: change to “described” not “described”
Reply: corrected
- MATLAB code, add where this code will be accessible? Github?
Reply: Thank you for pointing this out. The MATLAB code will be accessible from the corresponding author and later at Github. This has now been addressed in the method section (line 276-277):
The code is accessible from the corresponding author.
- Introduce acronyms when first used, including d for day.
Reply: Corrected.
- Figure 2 is confusing, why the dip?
Reply: We are not sure what you are referring to in Figure 2. If you meant the dip in Figure 4, we can respond as follows:
The reason is due to activity in the surrounding tissue. This activity will cross-irradiate, especially, in small tumors. Therefore, single cells and small clusters will be sterilized. However, when tumor size increases, this cross-irradiation will decrease, thereby, reducing TCP. Further increase in tumor size will, in contrast, increase self-irradiation, resulting in increased TCP. This is the reason for the particular shape of the curves. We have now added a short description into the manuscript (line 335-338):
However, if the activity concentration is high in surrounding tissue, the cross-irradiation from the normal tissue can sterilize the smallest tumors. Figure 4B illustrates this fact for a TNC of 10, i.e. TCP increase for decreased tumor sizes (< 10-6 cm-3).
- Table 3: is the mean size of mets an average of each met individually or in of all mets combined? Also, if there is a mean, please do add an SD value.
Reply: Thank you for this remark. The distribution will become log-normal distributed, the median and the 95% CI are presented (min and max value shouldn’t be used, since it is a stochastic process, with no well-defined max value). At line 313-314 the following sentence has been added:
For the median size of detectable metastases, the 95% confidence interval is presented in parentheses.
- Table 4: for Ac-225, why is there an Gy5in each column rather than a Gy written down?
Reply: We use the methodology as for references 6 to 8, indicating that a weighting factor of five was used in the calculation of the required absorbed dose.
In the table legend the following has been added (line 373-374): “For 225Ac, a weight of 5 was used in the calculation of the required absorbed dose; indicated with the unit “Gy5”.
- Add in methods or discussion what percentage binding is assumed of PSMA, which PSMA peptide is being used for the assumptions, what the assumed specific activities are, and whether an equal affinity is assumed for the target regardless of which chelator or radionuclide is used (this won’t always be the same).
Reply: In this study it was assumed that all radiopharmaceuticals had equal affinity and uptake in tumor tissue.
The following has been added to the method in paragraph 2.3 (line 222-223):
Furthermore, it was assumed that all radiopharmaceuticals had the same tumor biokinetic and had an effective half-life (Teff) of 51 h [3].
Reviewer 2 Report
The article by Bernhardt and co-workers describes a Monte Carlo model used for radionuclide therapy of metastatic prostate cancer and calculates the doses required for an array of therapeutic radionuclides including I-131, Lu-177 Y-90, Ac-225 as well as the emerging Tb-161 to control metastatic disease. The calculations are based on a Monte Carlo simulation for metastasis of prostate cancer developed by Bernhardt et al. Based on this model, the tumor cure probability is calculated for the above-mentioned radionuclides. The calculations show that Tb-161 might be more effective than currently used radionuclides for targeted radioligand therapy of (metastatic) prostate cancer.
This is a highly interesting contribution that perfectly matches the scope of Cancers. The study provides important insights into the theoretical aspects of radioligand therapy of prostate cancer and metastases providing useful information for practical implications to the clinic. The study is well designed, the manuscript is well-written and appropriate in length. Altogether, the manuscript is of high importance and well-suited for publication in Cancers.
However, here are some suggestions and open questions, which should be addressed prior to acceptance for publication:
Major
- The authors use a weighting factor of 5 for the alpha-emitters. Can the authors please explain why they used a factor of 5 and not 20 as one would expect?
- Can the authors please comment in more detail why a high metastatic formation rate results in smaller tumor volumes and vice versa?
- Do the authors have an explanation for the decreased tumor cure probability for Ac-225 and Tb-161 at about 10-7and 10-6 cm3 of metastasis volume, respectively (see Figure 4)?
- Have the alpha-emitting daughter nuclides of Ac-225 been included in the calculations or only the particles from Ac-225 decay?
- It is suggested to also include the beta-emitter Cu-67 in the study, which is now also available in large quantities and high purity. A side-by-side comparison with Lu-177 and Tb-161 would certainly be of high interest to the readers.
- The paper Rosar et al "New insights in the paradigm of upregulation of humoral PSMA expression .." Eur J Nucl Med Mol Imaging 2020 could be well cited in row 291 of the manuscript.
Minor
- Table 2. What does nt in the term KeV/nt mean?
- Equation 4: What does the term cT/TNC stand for?
- Line 148: Not complete sentence. Please rephrase.
- Line 151: Typo, equation 8 does not exist.
- Line 152: Replace were with was.
- Line 201: Please insert reference to Figure 4.
- Please check references for consistency. For example, some journal titles are abbreviated, some are given as full names, etc. For example, compare ref. 25 and 3.
Author Response
The article by Bernhardt and co-workers describes a Monte Carlo model used for radionuclide therapy of metastatic prostate cancer and calculates the doses required for an array of therapeutic radionuclides including I-131, Lu-177 Y-90, Ac-225 as well as the emerging Tb-161 to control metastatic disease. The calculations are based on a Monte Carlo simulation for metastasis of prostate cancer developed by Bernhardt et al. Based on this model, the tumor cure probability is calculated for the above-mentioned radionuclides. The calculations show that Tb-161 might be more effective than currently used radionuclides for targeted radioligand therapy of (metastatic) prostate cancer.
This is a highly interesting contribution that perfectly matches the scope of Cancers. The study provides important insights into the theoretical aspects of radioligand therapy of prostate cancer and metastases providing useful information for practical implications to the clinic. The study is well designed, the manuscript is well-written and appropriate in length. Altogether, the manuscript is of high importance and well-suited for publication in Cancers.
However, here are some suggestions and open questions, which should be addressed prior to acceptance for publication:
Reply: Thank you for your interest in our study. We have done our best to address your valuable remarks below.
Major
- The authors use a weighting factor of 5 for the alpha-emitters. Can the authors please explain why they used a factor of 5 and not 20 as one would expect?
Reply: The weight of 20 has been determined/estimated for radiation protection purposes, involves factors such as cell death and estimates late effects, such as induction of cancers. This value should be used only for radiation protection purposes and, when used, one should use its SI unit, Sv. This should not be applied to the therapeutic situation. The best estimate of the enhanced cell kill effect per Gy is around 5, and this value is often used in the literature. We have three references that use this value, however, they use Sv as unit, which we think is not correct. As a result, we made a note about this in Table 1. We indicate that a weighting factor of 5 was used by using the unit Gy5. Unfortunately, there is no nomenclature/consensus in this regard.
We have added the following to the caption of Table 4 (line 373-374):
For 225Ac, a weight of 5 was used in the calculation of the required absorbed dose; indicated with the unit “Gy5”.
- Can the authors please comment in more detail why a high metastatic formation rate results in smaller tumor volumes and vice versa?
Reply. The key here is that we specify a detectable tumor size. With a high metastatic formation rate, the time interval between all created metastases is much less than for a low metastatic formation rate. When metastases have grown to a size where it is detectable, the short time interval between the created metastases will cause what one detects e.g. 20 small tumors within a short time interval. These tumors do not have the time to become large. For the situation with low metastatic formation rate, the time interval between the formatted metastases are much longer and, therefore, the tumors can become larger when one has detected e.g. 20 tumors. In Table 3, it can be seen that it takes 900 days from 1 metastasis detected until 20 are detected with the low metastatic formation rate, ,that is, the first detected tumors have 900 days to grow. while it is only 340 days for the situation with high metastatic formation rate.
- Do the authors have an explanation for the decreased tumor cure probability for Ac-225 and Tb-161 at about 10-7and 10-6cm3of metastasis volume, respectively (see Figure 4)?
Reply: Thank you for noticing this. The reason is due to activity in the surrounding tissue. This activity will cross-irradiate, especially that of small tumors. Therefore, single cells and small clusters will be sterilized. When tumor size increases this cross-irradiation will decrease, thereby, reducing TCP. Further increase in tumor size, however, will also increase the self-irradiation, resulting in increased TCP. This is the reason for the particular shape of the curves. We added a short description into the manuscript (line 335-338):
“However, if the activity concentration is high in surrounding tissue, the cross-irradiation from the normal tissue can sterilize the smallest tumors. Figure 4 demonstrates this fact for a TNC of 10, i.e. TCP increase for decreased tumor sizes (< 10-6 cm-3).”
- Have the alpha-emitting daughter nuclides of Ac-225 been included in the calculations or only the particles from Ac-225 decay?
Reply: All alpha particles are included. It is described in the methods section, paragraph 2.3 (line 220-222).
- It is suggested to also include the beta-emitter Cu-67 in the study, which is now also available in large quantities and high purity. A side-by-side comparison with Lu-177 and Tb-161 would certainly be of high interest to the readers.
Reply: Thank you for the suggestion. We will certainly include Cu-67 in an upcoming study and perform an extension of our approach described here. This study, however, focuses on radionuclides that have been used preclinically/clinically for targeted PSMA therapy.
- The paper Rosar et al "New insights in the paradigm of upregulation of humoral PSMA expression .." Eur J Nucl Med Mol Imaging 2020 could be well cited in row 291 of the manuscript.
Reply: Thank you for notifying us of this most interesting paper. We will include it in our future study– studying different strategies to optimize MCP.
Minor
- Table 2. What does nt in the term KeV/nt mean?
Reply: We have changed nt to decay.
- Equation 4: What does the term cT/TNC stand for?
Reply: Apologies: we omitted to define the term. We have now added (line 238-239): …CT is the tumor activity concentration, TNC is the tumor-to-normal-tissue activity concentration ratio,…
- Line 148: Not complete sentence. Please rephrase.
Reply: The sentence has been corrected.
- Line 151: Typo, equation 8 does not exist.
Reply: Thank you. Corrected to equation 6.
- Line 152: Replace were with was.
Reply: Corrected.
- Line 201: Please insert reference to Figure 4.
Reply: Thanks. Now inserted.
- Please check references for consistency. For example, some journal titles are abbreviated, some are given as full names, etc. For example, compare ref. 25 and 3.
Reply: Now corrected.